# Beyond Context Limits: Subconscious Threads for Long-Horizon Reasoning

## Abstract

To break the limits of large language models (LLMs) in reasoning length that bottleneck reasoning accuracy and efficiency, we propose Thread-2, a JSON-based reasoning framework that naturally supports recursive reasoning and context pruning without examples or prompting, and the Thread Inference Model (TIM), a family of LLMs trained for recursive and decompositional problem solving. By learning to solve complex problems with structured trajectories defined by Thread-2 with supervised fine-tuning and reinforcement learning on synthetic data, TIM supports virtually unlimited reasoning length and multi-hop tool calls within a single language model inference, overcoming output limits, positional embedding constraints, and GPU memory bottlenecks. Such performance is achieved by modeling natural language as reasoning trees measured by both length and depth instead of linear sequences. Thread-2 reasoning trees consist of tasks with thoughts, recursive subtasks, and conclusions based on the recursive reasoning concept proposed in (Schroeder et al., 2025). During LLM inference, we maintain a working memory that retains only the key/value states of the most relevant context tokens, selected by a rule-based subtask-pruning mechanism, enabling the reuse of positional embeddings and GPU memory pages throughout reasoning. Experimental results show that our system sustains high inference throughput, even when manipulating up to 90% of the KV cache in GPU memory. It also delivers accurate reasoning on mathematical tasks and handles information retrieval challenges that require long-horizon reasoning and multi-hop tool use.

## 1 Introduction

Large language models (LLMs) have emerged as versatile foundations for a wide range of AI applications, especially agents which handle complicated tasks including multi-hop reasoning and tool use. Their ability to generalize across various tasks with minimal fine-tuning has driven rapid innovation and broad adoption (Brown et al., 2020). However, the fundamental objective of language modeling, to generate unstructured token sequences (Bengio et al., 2003), imposes strict context window limits and makes fine-grained control over internal state difficult. As a result, these inherent constraints pose significant challenges for all state-of-the-art LLMs, notably their inability to maintain long-horizon reasoning trajectories and coordinate complex workflows, which hinders the development of robust, memory-intensive applications.

Neural networks generate natural language as a linear sequence. Recurrent neural networks (RNNs) (Mikolov et al., 2010; Luong et al., 2015; Gu & Dao, 2023) and Transformers (Vaswani et al., 2017; Yang et al., 2023) are constrained by token limits, hidden state sizes, and GPU-memory capacities. For example, standard deployments of Deepseek R1 (Guo et al., 2025) offer up to 128K tokens across inputs and outputs, but real-world applications often require reasoning over longer horizons, especially when LLMs are connected to arbitrary outputs from external tools. Specialized architectures like the Compressive Transformer(Rae et al., 2019) compress past activations into secondary memory buffers to extend context, but these approaches still face trade-offs between memory fidelity and computational efficiency.

To work around the working memory bottleneck, developers frequently partition complex workflows into multiple modules (namely multi-agent architecture), each backed by a separate model instance that is responsible for distinct subtasks. Multi-agent frameworks (Li et al., 2023; Hong

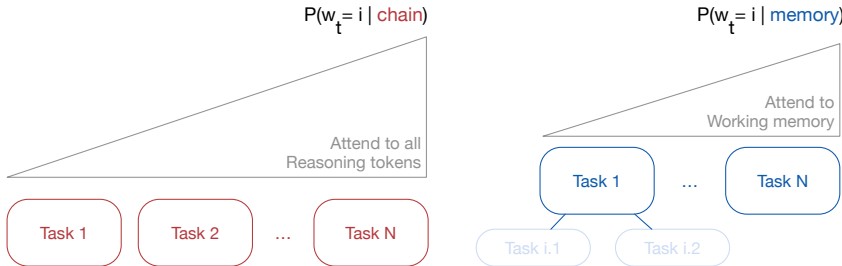

Figure 1: Latent information compression for all context tokens versus structural latent information compression focusing on the working memory enabled by parsing the reasoning trajectory.

et al., 2024; Wu et al., 2024) facilitate such workflows by dividing problems into tractable units. Domain-focused workflows demonstrate the power of agent societies in highly specialized settings with strong prior knowledge and well-defined scope. However, these multi-agent designs introduce significant overhead while dealing with more arbitrary tasks since agents do not inherently manage control flow or coordination, leaving developers to hand-craft context management, exception handling, and inter-agent communication. Moreover, integrating external tools further compounds complexity. Parameter generation, tool calling, and tool response processing are usually handled by different modules, inflating both development effort and runtime latency.

We believe that reasoning is not a linear process; it is recursively structured with inner dependencies, just like language (Aho & Ullman, 1972), hinted at by many real-life experiences. For example, in programming tasks, we often focus on the lines around the cursor, recall the inputs and outputs of the functions we have completed, and keep TODOs in mind. We no longer memorize all the details of a completed function, since our subconscious brain has flushed that information out of the working memory to help us focus on the current task. Inspired by this observation, we propose a new perspective to avoid the context and representation bottlenecks faced by traditional neural language models. We model a reasoning trajectory as a recursive tree of subtasks. While higher-level nodes in the tree receive tasks that require extensive multi-hop reasoning and tool use, the tree keeps decomposing complex instructions into simpler subtasks until reaching a leaf node, which represents a straightforward task that can be completed within one step. Our hypothesis is that processing an intermediate task does not have to attend to the subtasks of previous steps.

As shown in Figure 1, by pruning irrelevant subtasks, the model only focuses on a selective "working memory". Compared to transformers that model language as linear sequences of tokens, an LLM reasoning over pruned reasoning trees does not have to attend to all context tokens. Compared to recurrent architectures, attending to a dynamic working memory provides more flexible and richer contextual information than decoding with constrained latent representations (Ben-Kish et al., 2025). By decomposing extensive workloads into subtasks, the model can prune a significant number of context tokens and KV cache entries during reasoning. This enables virtually unlimited long-horizon reasoning while maintaining awareness of instructions and important context. As a result, the model achieves higher decoding throughput and reduced memory cost.

This paper reports our implementation of this idea, consisting of two major contributions. Firstly, we build the Thread Inference Model (TIM), a transformer-based LLM that recursively decomposes complex tasks, follows subtask instructions, and aggregates bottom-up subtask outputs. TIM also learns to appropriately use multiple external tools within subtasks to complete a complex workflow in a single language model inference call. By generating a highly structured reasoning trajectory, TIM can easily recognize decomposed subtasks, tool parameters, and the hierarchy of the recursion. Equally important, we designed Thread-2, a JSON-defined reasoning framework for TIM and an extension to Thread. The framework identifies the structure of reasoning during inference, helps the underlying inference system dynamically release the memory occupied by the KV states of subtasks that are no longer helpful, and reuse that memory in further inference. Powered by Thread-2 and a corresponding subtask pruning strategy, TIM achieves the following breakthroughs:

- Performs virtually unlimited long-horizon reasoning beyond output token limits
- Enables efficient single-model reasoning for complex tasks with higher decoding throughput and memory efficiency
- Unlocks the possibility to build agents in a most concise manner: giving TIM a toolkit, launching one model inference, and receiving agentic reasoning trajectory.

Figure 2: The pydantic class we use to create the JSON schema for constrained decoding.

## 2 THREAD INFERENCE MODEL (TIM)

We model reasoning trajectories as recursive subtask trees and train a transformer-based model to learn this structure. This section first introduces the improved thread structure we designed for reasoning, and then introduces our data synthesis and model training pipelines.

### 2.1 THREAD-2

In our design, the basic unit of reasoning is a `task`, consisting of a thinking process, an optional tool use, an optional subtask list and a conclusion. The roles of these fields are: (1) `thought`: contains a thinking process that catches the mistakes of previous steps, analyzes current progress, and plans the following steps. (2) `tooluse`: optionally call a specific tool by generating the input of the tool and encode the responses of the tool after receiving them. (3) `subtasks`: optionally spawns subtasks if the current task needs multi-step reasoning. The reasoning details of the spawned subtasks will be hidden from the next step for efficient and concise context management. (4) `conclusion`: processes tool results, aggregates the conclusion of the subtask list in the current step, and describes the result of the current task, which is informative enough to support future reasoning tasks.

All `tasks` in the reasoning tree share the same schema. Compared to the initial Thread reasoning framework (Schroeder et al., 2025), Thread-2 makes several improvements. Firstly, Thread does not pass the instruction of a higher-level task to subtasks, each subtask needs a copy of the system message to realize recursive subtask spawning. This setting introduces inefficiency in decoding and a potential information gap, since the subtask instruction might not cover all necessary inputs. If we naively append all descriptions of higher-level task to the subtask instruction with careful prompt engineering, even large models can still be confused and work the wrong instruction. Thread-2 fixes this issue by accessing the working memory, containing the system prompt, user input, and all tasks that are not pruned. Conversely, with Thread-2, the language model conducts end-to-end inference, finishing the reasoning with only one language model call.

**Subtask pruning.** Similar to Thread, subtasks and the recursion hierarchy can be easily extracted. Therefore, we can reduce the complexity of the reasoning context with a rule-based subtask pruning mechanism, without using an external summarization model or agent history memory. Ideally, we believe that processing the current task only needs to read the thoughts and conclusions of previous tasks at the same or higher level, and can safely ignore pervious subtask lists in lower levels. However, the model often needs more redundancy and flexibility to deliver a more accurate reasoning result. As a result, we prune subtasks through a subtask stack with a fixed size. When a subtask list is completed, we add this list to the stack. If the stack size is larger than the threshold, we pop the earliest subtask list and prune it from the working memory. In practice, we set the threshold among $\{0, 1, 2\}$. At the subtask level, this mechanism is similar to StreamLLM (Xiao et al., 2023), but with more attention sinks dynamically decided by the subtask recursion structure.

**Structured generation.** Instead of defining special tokens $\phi$ and $\psi$ as structure operators with a few-shot task-specific prompting and multiple LLM API calls, the Thread-2 reasoning process can be efficiently decoded as a JSON dictionary with popular inference runtimes (Kwon et al., 2023; Zheng et al., 2024) with constrained decoding engines (Willard & Louf, 2023; Dong et al., 2024).

In practice, we perform JSON decoding using the schemas shown in Figure 2, demonstrated with example search and web reading tools. Note that multiple tool calls can be handled with one decoding pass. Traditionally, a reasoning process with multiple tool calls is mainly based on the message

list design. Tool responses are appended to the message list as user input and the entire message list will be resubmitted to the LLM serving system. Although most message entries can be cached so that the KV states of those tokens do not have to be recomputed, the overhead of state caching, network transition, and cache matching for each tool call can significantly decrease overall generation throughput. With proprietary LLM services, developers have to pay for the cached tokens multiple times. For example, if a process requires 20 tool calls, the developer might be charged for their initial input tokens 20 times.

During generation, TIM extracts the previous parameters as a dictionary when it outputs the `tool_result` keyword. Instead of sending the parameters to the developer or another module, and requiring manually appending the tool responses into the message list and resubmitting to the LLM, TIM waits until receiving tool responses as dumped JSON dictionary strings in the reasoning runtime and extends its KV cache by encoding them as batches of new input tokens. This mechanism allows for the use of multiple tools with just one language model call, avoiding the overhead of network latency and caching and retrieval of multiple tools.

## 2.2 TRAINING

In this study, we post-train a small open-source model with a small, synthetic corpus as a proof of concept. The goal of this preliminary training is to prove our main hypothesis about our method: 1. Subtask pruning does not harm reasoning accuracy, and 2. Intensive management of the KV caches incurs no additional computational overhead.

**Supervised fine-tuning.** To produce a model that natively generates Thread-2 reasoning structures without a heavy prompt, we created a synthetic training set and trained a `Qwen3-8b` model (Bai et al., 2023). We constructed a set of questions by taking 20k openr1-math-220k questions (Hugging Face, 2025), 20k research questions (Rosset et al., 2024), and 6k ToolBench questions (Guo et al., 2024). We then assign the available tools for different types of question. For math questions, we simply prohibited the use of tools. For research questions, we allow for a search tool and a webpage reading tool. For each question from the benchmark, we synthesize its tool I/O schemas according to the associated example input and output. After constructing the question-tool pairs, we send them to a collection of large language models and generate JSON dictionaries by replacing the tool-related schema shown in Figure 2.

**Reinforcement learning.** We also carry out reinforcement learning for the fine-tuned model on the remaining questions from openR1-math-220k (Hugging Face, 2025) with GRPO (Shao et al., 2024; Sheng et al., 2024). We continue to enforce the JSON structure during sampling and provide the reward by comparing predicted and annotated answers. We noticed that although the data set we used for training supervised learning is noisy and we conduct rollout with a constrained format, GRPO can still improve the performance of the fine-tuned model.

## 3 IMPLEMENTATION OF THE INFERENCE RUNTIME

TIM's structured output offers new opportunities to enhance reasoning performance and accuracy. However, this novel reasoning format also poses new challenges for deployment. To fully harness TIM's potential and address the deployment obstacles presented by the Thread-2 reasoning framework, we designed an inference algorithm for efficient subtask pruning, tool response extending, and batching.

The key difference between TIM and traditional agents lies in how they utilize input and output windows, which introduces practical challenges for TIM's deployment. Traditional agents progressively update message lists, and the underlying LLM encodes them as part of the input sequence. In contrast, TIM executes the entire reasoning process in the output window. This approach is difficult to realize in practice, as many language models have much stricter output window limits compared to inputs. For instance, Qwen 2.5 supports 128k tokens for input but only 32k for output. To enable long-horizon reasoning that exceeds the output limit, the inference runtime must support reuse of both GPU memory and positional embeddings for output generation.

### 3.1 SUBTASK PRUNING

Subtask pruning is essential to efficiently implement TIM and sustain long-term reasoning. The core idea is that, at any moment, the model only needs the outputs of prior tasks at the same level of abstraction; it can safely discard the internal details of their subtasks. The thought experiment in Zhao & Song (2000) captures this methodology:

> "*How to put an elephant in a refrigerator?*
> *Three steps. Open the door, put the elephant in, then close the door.*"

To implement this principle, we design a pruning buffer, a stack that temporarily caches a small set of prunable subtasks, retaining just enough redundancy to ensure lossless information flow. The subtask pruning process is shown in Figure 3. While TIM decodes within a task, it dynamically evicts the KV states of tokens belonging to completed subtasks from GPU memory. Such fine-grained memory manipulation could occur inside the forward pass, but in practice that approach imposes extra computation and latency compared to computing attention against a longer KV cache.

To minimize the overhead of GPU memory management, we process the KV cache and prune subtask tokens before inference with paged-attention (Kwon et al., 2023). For the dynamic subtask pruning mechanism enabled by structured generation, we set the page size to 1 since each request in the same batch requires different pruning. The batched pruning is implemented with Triton (Triton, 2021), and inference with page size as 1 is accelerated by FlashInfer (Ye et al., 2025). Given a token sequence $S$ and the current KV cache $H$ before pruning:

$$S = [t_1, t_1^1, t_1^2, t_2, t_2^1, x_k], \ H = [h_1, h_1^1, h_1^2, h_2, h_2^1]$$

where $x_k$ is the new input token pending encoding, $t_i^j$ stands for tokens in the $j$-th subtask of task $i$. The corresponding hidden states will be represented by $h_i^j$ and $h_k$. Assume that the model predicts the next token $x_{k+1}$, and following the pruning rule, we remove $t_1^1, t_1^2$ from the cache before decoding. The remaining sequence and hidden states after pruning and before decoding will be

$$S' = [t_1, t_2, t_2^1, x_k], \ H' = [h_1, h_2, h_2^1]$$

We could simply use $H'$ as the new KV cache and continue the decoding. However, although the memory pages occupied by the pruned tokens can be recycled thus improving memory efficiency, TIM cannot decode more tokens beyond the output limit since the encoded positional embeddings are not re-used. As a result, we need to re-encode, or extend all tokens after the pruned subtasks to re-assign positional embeddings:

$$(h_2', h_{2.1}', h_k; x_{k+1}) = f_{extend}(t_2, t_{2.1}, x_k \mid h_1), \ H^* = [h_1, h_2', h_{2.1}', h_k] \tag{1}$$

where $x_{k+1}$ is the next predicted token and $H^*$ stands for the updated KV cache, with task 1.1 and 1.2 pruned, for the next decoding step. Although the re-encoding process increased the amount of computation, the new tokens are encoded in parallel by GPU kernels. Therefore, the overall throughput will not be significantly impacted. In addition, the positional embeddings of the pruned tokens are reused, and those previously occupied by the extended tokens are recycled for further reasoning. With appropriate subtask decomposition, TIM can reuse both GPU memory and positional embeddings iteratively in the output window without running out of those resources, enabling long-horizon reasoning beyond the predefined output limit.

### 3.2 END-TO-END MULTI-HOP TOOL USE

Tool use and multi-agent frameworks often incur excessive token costs due to repetitive prefilling. In autoregressive LLM generation, each inference involves two stages: prefilling and decoding. Prefilling encodes all input tokens at once, storing their hidden states in the KV cache, and generating the first output token. After prefilling, only new tokens are processed for subsequent predictions. This process is called 'extending' when some prefix is cached. Most LLM APIs accept inputs as a message list representing a multi-turn interaction. For every new user turn, the latest message is appended and the entire message list is sent to the LLM, repeatedly re-sending most of the context. To optimize computation, inference engines cache hidden states for previous tokens, so only the new tokens from the latest user input are encoded and added to the KV cache.

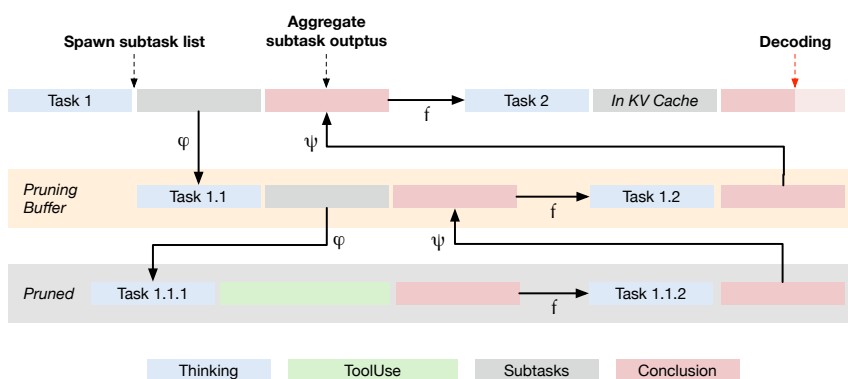

Figure 3: While TIM is decoding the conclusion of task 2, tokens in task 1.1.1 and 1.1.2, including the enclosed tool call and response have been pruned from the KV-cache. $\psi$ stands for subtask spawning, $\phi$ is subtask aggregation, and $f$ appends a step in the current task list.

Despite caching, token retrieval and network transmission introduce extra overhead. More importantly, commercial LLM APIs typically charge for "encoding" cached tokens, so developers pay for the same tokens multiple times, even if they are not actually re-encoded. At minimum, the redundant cost scales with the number of "extend" requests. In some multi-agent architectures, each reasoning step is treated as a new request, resulting in approximately $O(n^2)$ cost complexity, where $n$ is the number of reasoning steps.

Our design addresses this problem by initiating tool calls directly within the runtime, rather than sending tool parameters back to the client. As illustrated in Figure **??**, this approach significantly reduces inter-module communication, streamlining agent development and deployment. TIM's structured generation makes this process seamless: whenever TIM outputs `tool_result:`", the inference system can easily extract relevant parameters from the `parameters:`" field, loads them as a JSON object, and forwards the request to the external tool (e.g., on an MCP server), then appends the tool's response to the ongoing reasoning process. Crucially, each token in the reasoning chain is forwarded by the model only once, eliminating redundant token transmission and minimizing communication overhead. This design also supports typical chatbot applications. After generating each response, the model server initiates a tool call to deliver the response to the user and collect subsequent user inputs.

## 4 EXPERIMENTS

In this section, we report the benchmark results of our model on reasoning and research tasks. The results of our experiments present the following observations. Firstly, maintaining a working memory instead of computing the attention weights to all context tokens does not hurt the reasoning accuracy. Furthermore, pruning irrelevant context can even improve the reasoning accuracy and reduce hallucinations for language models. Secondly, TIM maintains high throughput despite intensive memory access and manipulation.

### 4.1 REASONING

We evaluated TIM models on MATH500, MMLU-STEM500, AMC 2022, AMC 2023, AIME 2024, and GPQADiamond to assess their STEM knowledge and reasoning abilities. The preliminary results are shown in Table 1. We compare TIM's reasoning accuracy across different serving infrastructures. When hosted with SGLang Zheng et al. (2024), TIM produces structured output following the schema in Figure 2, without subtask pruning. In contrast, TIM + Pruning refers to TIM served with our custom inference system, which introduces subtask pruning and memory management during decoding.

The results demonstrate that subtask pruning for TIM does not degrade overall performance. In fact, retaining only the most relevant information in the KV cache rather than storing all reasoning tokens

| Task | TIM-8b + SGLang | TIM-8b + Pruning | | | |
|------|-----------------|------------------|---|---|---|
| | Accuracy | Accuracy | Max Cache | Output Len. | KV Pruned (%) |
| MATH500 | 69.6 | 69.0 | 1569.2 | 3362.2 | 53.3 |
| MMLUSTEM500 | 88.4 | 87.6 | 1330.9 | 2747.0 | 51.6 |
| AMC 2022 | 60.5 | 60.5 | 2203.9 | 5131.7 | 57.1 |
| AMC 2023 | 80.0 | 80.0 | 1876.5 | 4547.4 | 58.7 |
| AIME 2024 | 40.0 | 46.7 | 3218.6 | 8974.7 | 64.1 |
| GPQADiamond | 44.9 | 48.5 | 1712.9 | 3742.6 | 54.2 |

Table 1: Evaluation results of TIM models served on different infrastructures. Max Cache stands for the maximal cache usage achieved during the entire generation flow. Output Len. stands for the number of the actual output tokens.

improves the TIM model's performance on many tasks. We report the maximum KV cache length observed during generation. Across all tasks, TIM achieves the reported performance while using less than half the cache slots required for the full output sequence. Notably, the peak KV cache length typically occurs only once during generation. For most other steps, the actual KV cache size is even smaller. Thus, the reported KV Pruned value represents only a lower bound on the memory savings enabled by TIM and subtask pruning.

## 4.2 RESEARCH

Large language models augmented with external knowledge typically need to generate search queries for information retrieval tools. In conventional implementations, query generation, tool invocation, and aggregation of tool responses are orchestrated by agentic workflows (Alibaba, 2024). In our experiments, TIM streamlines this process by efficiently extracting tool parameters, invoking the necessary tools, and appending the raw tool responses directly to the output sequence. Therefore, developers are no longer required to implement complex agent workflows. Multi-hop tool use is handled by TIM as a seamless, end-to-end LLM API call.

Following this design principle, we evaluated TIM models on agentic research tasks without relying on any agent framework or complex prompting strategies. We used two benchmarks, BrowseComp (Wei et al., 2025) and Datacommons QA (Guha et al., 2023; Schroeder et al., 2025), both requiring multi-hop information retrieval, processing of tool responses, and reasoning.

**Datacommons QA.** Following the experimental setup in Schroeder et al. (2025), we provide the model with a search tool to interact with Datacommons and evaluate its performance on 140 benchmark questions. Our primary baseline, Thread, uses a task-specific prompt with over 4,000 tokens and includes two detailed examples for Datacommons queries and APIs. Other baseline methods also rely on few-shot prompting with hand-crafted examples tailored to the Datacommons task (Khot et al., 2023; Shinn et al., 2023). In contrast, TIM only requires a concise system message and essential information about the tool, including tool description, input parameters, and the output format. We note that the model was not trained on the Datacommons tool utilized and leave exploration of improved performance through fine-tuning on specific tool usage for future experiments. The experimental results are summarized in Table 2.

| Method | Reflection | NLEP+ReACT | DecomP | THREAD | TIM |
|--------|-----------|------------|--------|--------|-----|
| Accuracy | 24.3 | 27.1 | 57.9 | 67.9 | 67.9 |

Table 2: Performance of different methods on the Datacommons QA benchmark. TIM is the only method that does not require task-specific few-shot prompting.

The reported performance shows that the TIM model generalizes well to novel tasks not encountered during training. Compared to baseline methods, TIM offers greater efficiency in three key areas. First, it eliminates the need for carefully crafted few-shot examples and task-specific prompts. A simple system message is sufficient for strong performance. Second, bypassing the 4,000-token prompt substantially reduces computational overhead during generation. Finally, developers are no

| Model | Deepseek-R1 | GPT-4o | TIM-large | TIM-8b |
|---|---|---|---|---|
| Paradigm | ReACT | Browsing | Browsing | Browsing |
| Success (%) | 9.5 | 1.9 | 7.8 | 2.3 |

Table 3: Success rates of LLMs without post training for Browsing under different paradigm.

longer required to develop bespoke logic for tool response handling, given that TIM automatically processes tool responses upon subtask completion and removal from the pruning buffer.

**Browsecomp.** Browsecomp is a challenging benchmark for deep research agents (OpenAI, 2025; Wei et al., 2025). Answering questions in this benchmark requires decomposing the input, using tools to filter and retrieve relevant information from the Internet, sometimes drilling down into specific webpage details, and validating findings against given conditions. Traditionally, such tasks require agent-based systems capable of managing long reasoning chains and aggregating multiple tool responses, often relying on models post-trained with search tools for related tasks (Li et al., 2025).

We constructed a system that enables GPT-4.1 to generate the JSON structures we designed for TIM with a generic system prompt, `TIM-large`. TIM-large, is more capable than our smallermodel 8b parameter model; however, it is less efficient as it is not served on our local inference system. We use TIM-large to validate the performance of the reasoning ability of TIM's JSON-based thread-2 pipeline. Similar to our experiments in Datacommons QA, we do not have an agent framework to manage the contexts for the model. Instead, we implemented the subtask pruning mechanism to ensure context efficiency.

The experiment results for frontier large language models without post-training on deep research tasks or tools are presented in Table 3. Without any agent design, Tim-large significantly outperforms GPT-4o with browsing capabilities and achieves performance comparable to the ReACT agent built on Deepseek R1, a strong reasoning model. These findings support our hypothesis: a model that autonomously manages its own context by recursively decomposing subtasks and pruning its working memory can match the performance of agents in more complex implementations. In particular, even TIM-8b, when decomposing research tasks, outperforms GPT-4o in the end-to-end browsing setting.

### 4.3 Efficiency and Scalability

TIM's ability to dynamically prune subtasks and maintain a working memory with less than 50% context tokens brings new possibilities to improve the throughput and memory efficiency of LLM serving systems. In the experiments to test the efficiency of TIM, we focus on two questions: 1. Does the intensive KV cache manipulation bring additional computation overheads, and 2. With a smaller working memory, can we decode bigger batches than without context pruning.

**Memory management overhead.** Motivated by the observation that pruning the KV cache should reduce the computational cost of the attention mechanism, we conducted experiments using native Huggingface and PyTorch implementations. However, we found that the overhead introduced by memory management actually outweighs the savings from a shorter KV cache. With a batch size of 1, the standard decoding implemented by the plain Huggingface transformers package with eager attention achieved 22 tokens per second, while decoding with KV cache pruning dropped to 18 tokens per second, a nearly 20% decrease in throughput. These preliminary results suggest that, in practice, memory management for cache pruning can be less efficient than simply computing attention over longer contexts.

**Improved throughput with Pruning.** As shown above, there is a trade-off between context pruning and attention computation. While pruning the context can accelerate attention computation, it also introduces additional memory overhead. We find that TIM strikes an effective balance. Despite the demands of structural checks and frequent memory access, it delivers improved throughput at the same batch size.

On AIME 2024 challenges, we evaluated different pruning buffer sizes and compared the throughput for each configuration with a batch size of 30. The results are shown in Figure 4a. When maintaining

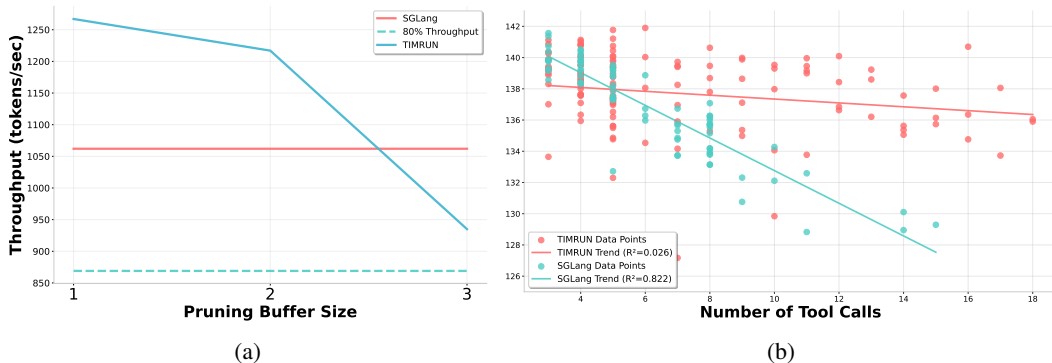

(a)     (b)

Figure 4: Throughputs of TIM model under different settings compared to SGLang. (a) analyzes the trade-off between memory management and KV cache size We found that setting the size of pruning cache to 2 achieves both good reasoning accuracy and inference throughput. (b) compares the throughput of TIM (red) and SGLang (blue) in multi-turn tool use tasks.

fewer than two subtask lists in the pruning buffer, the KV cache in GPU memory remains sufficiently compact. In this setting, the time saved on attention computation through subtask pruning outweighs the additional memory access overhead, leading to higher throughput than the baseline system. However, as the size of the pruning buffer increases and fewer subtasks are pruned, the system incurs more memory management overhead without sufficient computational savings.

We also provide the 80% throughput line in the plot to represent the result we obtained with memory access during the forward phase of the model inference. Overall, the results show that the TIM system outperforms both naive memory operations and the strong SGLang baseline.

**More efficient tool use.** Table 1 indicates that TIM can invoke custom tools end-to-end directly from the runtime, bypassing the client or developer. This approach offers several advantages for development simplicity and inference scalability. By calling tools and encoding results within the runtime, multiple overheads are avoided. First, network transmission latency is reduced since tool parameters do not need to be sent between the runtime and the client. Second, the runtime eliminates the need to cache tokens and manage their associated states. Most importantly, TIM's subtask pruning mechanism further enhances inference efficiency by removing prior tool responses together with completed subtasks.

Experiment results shown in Figure 4b support our hypothesis. We evaluated the TIM-8b model served on both SGLang and our inference implmenetation with subtask pruning using BrowseComp tasks. We analyze the relationship between average throughput and the number of tool calls. As expected, SGLang's throughput drops rapidly as the number of tool calls increases, due to the growing complexity of incremental context and token cache from reasoning steps and tool responses. In contrast, TIM with pruning maintains relatively stable throughput even as tool usage scales, thanks to its automatic context management mechanism. This enables the TIM-8b model to achieve strong performance on the BrowseComp benchmark, without any agent framework or task-specific post-training. With subtask pruning, TIM supports more than 30 tool calls within a single inference.

## 5 CONCLUSION

In this work, we introduce a co-designed system consisting of a large language model, TIM, and its dedicated reasoning framework, Thread-2. TIM is trained to decompose complex tasks into simpler subtasks and reason over a recursive JSON structure. With the structured reasoning trajectory design, TIM enables efficient subtask pruning, batching, and end-to-end tool integration. Our experiments show that generating a more concise KV cache not only increases inference throughput, but also enhances performance on certain tasks by helping the model focus on relevant context. In agentic benchmarks, TIM without explicit agent-specific design matches the performance of strong baselines that rely on more complex agent frameworks and task-specific post-training. Overall, the combination of Thread-2 and TIM delivers strong reasoning ability, more efficient inference and tool use, and greater flexibility and scalability for agentic tasks.

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
