# OpenReview forum: "Beyond Context Limits: Subconscious Threads for Long-Horizon Reasoning"
_ICLR.cc/2026/Conference — Submitted to ICLR 2026_

### Official Review · Reviewer_QfX4 · 2025-10-31

**Soundness:** 2
**Presentation:** 2
**Contribution:** 2
**Rating:** 4
**Confidence:** 3

**Summary:**

The authors propose TIM, an LLM-based system that recursively decomposes and reasons for solving complex problems. TIM divides a problem into tasks, and each task is structured as a JSON dictionary containing four components: Thought, ToolUse, Subtasks, and Conclusion. To efficiently manage information for nested subtasks, TIM’s Thread-2 mechanism uses KV-cache–based structured pruning along with a task stack. TIM can handle multiple tool calls within a single model call, and tool outputs are returned in JSON format so they can be directly used in subsequent steps. Experiments on challenging reasoning and multi-hop tool-use benchmarks such as MATH500, MMLUSTEM500, and AMC show that TIM achieves performance comparable to or better than existing agent-based systems.

**Strengths:**

-By using subtask pruning and KV-cache management, TIM optimizes GPU memory usage compared to traditional message-list–based approaches.
- The reasoning process can be explicitly represented in a JSON dictionary format, without requiring special tokens or complex prompt templates.
- While the idea of breaking complex problems into smaller tasks has been used in prior systems (e.g., Gemini Deep Research), those approaches typically divide tasks linearly (i.e., step-by-step in sequence). In contrast, TIM recursively decomposes tasks, enabling more structured and logically coherent reasoning.
-Because the system achieves strong performance with only a simple system prompt, the burden of designing complex prompts is significantly reduced.
-TIM-8B not only outperforms GPT-4o on BrowseComp, but also demonstrates consistent performance gains as more tool calls are utilized, highlighting the benefit of its multi-step reasoning design.

**Weaknesses:**

-Performance improvement is not significant.
-Insufficient breadth in experimental comparisons.
The datasets used in the experiments (MATH, MMLU-STEM, AMC, AIME, GPQA-Diamond) are not well suited for evaluating long-context reasoning. Most of these datasets feature short problem statements and do not require reasoning over extended context. Typically, long context tasks involve tens or hundreds of thousands of tokens. Therefore, the choice of datasets seems misaligned with the paper’s stated goal. It would have been more appropriate to include established long-context benchmarks (e.g., LongBench, ZeroSCROLLS, InfiniteBench, BABILong).
-Ablation studies are incomplete.
Figure 4 presents changes in throughput as a function of pruning buffer size, but does not report the corresponding accuracy results. Without examining accuracy trade-offs, it is unclear how pruning affects model robustness. For example, how does accuracy change when the pruning buffer size is 1, 2, or 3? Including an ablation table comparing buffer size vs. accuracy vs. throughput would provide a more comprehensive evaluation.

**Questions:**

-There are many places which need more explanations or clarifications. Here is a list of concerns and questions.
1.The paper does not provide a quantitative measure of problem complexity, making it unclear when and why a problem should be recursively decomposed.
2. There is no concrete algorithm or rule for determining how to optimally split tasks, such as criteria for the number of subtasks, depth of recursion, or balanced decomposition.
3. It is not discussed whether the system produces a consistent subtask tree for the same input. Due to the uncertainty nature of LLMs, the structure may vary across runs.
4.There is no direct metric to evaluate the quality of task decomposition. The paper lacks failure-case analysis or definitions of what constitutes an incorrect decomposition.
5. The framework does not specify a scheduling policy for execution order in complex hierarchical task trees, leaving task prioritization ambiguous.
6. For highly complex problems, the subtask pruning buffer may grow large, leading to increased GPU memory overhead and reduced throughput.
7. If the LLM produces syntactically invalid JSON, the entire reasoning pipeline may fail, and the paper does not provide a clear error detection or recovery mechanism.
8. -The subtask pruning method deletes the oldest entries when the stack is full, which risks removing tasks that may be needed later. A frequency- or relevance-based eviction strategy may be more appropriate.
9. Training on synthetic data may not fully capture the diversity and irregularity of real-world hierarchical task structures.
10. -The rule-based stack size limit (e.g., 0–2) may be sufficient for simple tasks but could lead to information loss or structural errors in deep, multi-step reasoning chains (e.g., beyond 5 layers).
11. The TIM family shows lower success rates than DeepSeek-R1, which uses a ReAct-style reasoning paradigm, suggesting potential limits in the proposed approach.
12. more explanation is needed to understand the extend (re-encode)
13. What happens if the problem is not recursively pruned, e.g., subtasks are lined up in a sequence.
14. “As illustrated in Figure ??” need to be fixed
15. In the subtask pruning process, the pruning threshold is set to one of {0, 1, 2}. Is this threshold determined heuristically, or does the model assign it automatically? What is the basis for choosing these specific values (0, 1, and 2), and how does the choice of threshold affect model performance?
16. In Section 4.2, the TIM-large pipeline uses GPT-4.1, while the baseline pipeline uses GPT-4o, which has weaker reasoning capabilities. This makes the comparison seem unfair.

---

### Official Review · Reviewer_RGs1 · 2025-11-01

**Soundness:** 2
**Presentation:** 2
**Contribution:** 2
**Rating:** 2
**Confidence:** 3

**Summary:**

The work introduces Thread-2, a JSON-based reasoning framework that organized reasoning as recursive trees of subtasks and allows dynamic pruning or irrelevant information for a clean working memory. The work also introduces Thread Inference Model (TIM), a model trained with SFT and RL to perform recursive reasoning and multi-hop tool use in a single LLM inference. Experiments show high inference throughput and accurate reasoning on mathematical tasks as well as tasks that require long-horizon reasoning and multi-hop tool use.

**Strengths:**

* The work introduces a new framework, Thread-2, that models reasoning as recursive trees of subtasks and allows dynamic pruning or irrelevant information for a clean working memory. This is a novel and interesting idea.
* The proposed TIM offers a training recipe for recursive reasoning and multi-hop tool use in a single LLM inference. This shows how model training can be adapted to the proposed inference framework.
* This work addresses a core problem in LLM-based reasoning, which is the limited context window along with memory scalability. Benchmarks on a variety of reasoning and tool use tasks show the effectiveness of the proposed method.

**Weaknesses:**

* The model is based on Qwen3-8B, yet performance is much lower than the baseline reported in the technical report [1]. Qwen3-8B achieves 76.0 in AIME 2024 (Table 17, Qwen3 technical report) but proposed models only achieve 40.0 (Table 1, the current work).
* TIM executes the custom tools at LLM inference backend and treats this as an advantage. However, this may not be the case for real-world applications, where the custom tools (e.g., computer use) may not be available at the inference backend. Furthermore, inference servers may not be connected to the internet or may not be suitable for executing LLM-generated code in a sandboxed environment colocated with the inference server.
* AIME 24 evaluation metric is not reported with averaging. Since AIME 24 only has 30 problems each, the results may be unstable. A common practice is to report the average score over multiple runs (e.g., Avg@32).
* No comparisons or discussions with adaptive reasoning methods that allow reasoning in parallel and can also save context by dropping irrelevant information from subtasks. For example, APR [2] only returns the necessary context for the next step to main thread, which is very similar to the proposed method.
* Writing is not clear. For example, the `MMLUSTEM500` is not spelled correctly as `MMLU-STEM500` in Table 1.
* Writing: Line 294 has a missing reference to a figure.

[1] Qwen3 Technical Report. Yang, et al. https://arxiv.org/abs/2505.09388

[2] Learning Adaptive Parallel Reasoning with Language Models. Pan, et al. https://arxiv.org/abs/2504.15466

**Questions:**

1. Can the authors clarify why such a large degradation occurs between the Qwen3-8B model and TIM? For example, is it due to the synthetic Thread-2 training data, architectural changes during inference, or pruning configuration?
2. In Fig. 4, the work claims to be able to achieve a higher throughput than the SGLang baseline in some cases. Does TIM achieve higher throughput than the SGLang baseline in the cases or settings in Table 1?
3. What is the relationship between Thread-2 and SGLang? Is Thread-2 a inference strategy implemented on top of SGLang, or is it a framework independent of the underlying inference engine?

---

### Official Review · Reviewer_uiag · 2025-11-01

**Soundness:** 2
**Presentation:** 3
**Contribution:** 3
**Rating:** 4
**Confidence:** 3

**Summary:**

This paper addresses the critical limitation of fixed context windows in Large Language Models (LLMs), which hinders their ability to perform long-horizon reasoning and complex, multi-step tasks. The authors propose a co-designed system consisting of Thread-2, a JSON-based framework for recursive reasoning, and the Thread Inference Model (TIM), an LLM fine-tuned to generate and follow this structured format.

The core idea is to model reasoning not as a linear sequence but as a recursive tree of tasks and subtasks. This structure allows for a novel inference-time optimization: a rule-based mechanism prunes completed subtasks from the KV cache, creating a compact "working memory". By dynamically managing the KV cache and re-assigning positional embeddings for subsequent tokens, the system can theoretically achieve "virtually unlimited" reasoning length, overcoming GPU memory bottlenecks and output token limits. This enables complex workflows, including multi-hop tool use, to be executed within a single, continuous model inference call. The authors validate their approach through experiments on mathematical reasoning and agentic research tasks, demonstrating sustained accuracy, improved inference throughput, and enhanced scalability compared to traditional methods.

**Strengths:**

The key innovation is not just the structured output but the direct manipulation of the KV cache based on this structure to enable dynamic context pruning.  This moves beyond passive context compression or simple retrieval to an active, structured memory management system.  The idea of enabling multi-hop tool use within a single inference call by handling tool I/O at the runtime level is a particularly elegant and significant departure from conventional, latency-prone agentic loops.  This approach has the potential to fundamentally change how complex agentic systems are designed and deployed.

**Weaknesses:**

1. Table 1 lacks necessary baselines, such as the accuracy of qwen3-8b.
2. The author mainly conducts evaluations in mathematics and web question-answering tasks. However, considering that the method in this paper is oriented towards tasks with extremely long contexts, the author should evaluate it on tasks that require long contexts, such as processing extremely long documents and SWE-bench.

**Questions:**

Please refer to Weaknesses.

---

### Official Review · Reviewer_jC79 · 2025-11-01

**Soundness:** 2
**Presentation:** 3
**Contribution:** 2
**Rating:** 6
**Confidence:** 4

**Summary:**

The paper introduces the Thread Inference Model (TIM), a novel family of large language models designed to overcome context limitations that hinder reasoning accuracy and efficiency. TIM employs a recursive and decompositional problem-solving approach, coupled with a context pruning mechanism, enabling long-horizon structured reasoning beyond traditional context constraints. This structure-aware context pruning allows TIM to support virtually unlimited working memory and multi-hop tool calls within a single inference, addressing challenges such as output limits, positional-embedding constraints, and GPU-memory bottlenecks. The model represents natural language as reasoning trees, consisting of tasks, thoughts, recursive subtasks, and conclusions, rather than linear sequences. During generation, TIM maintains a working memory that retains only the key-value states of the most relevant context tokens, selected by a rule-based subtask-pruning mechanism, facilitating the reuse of positional embeddings and GPU memory pages throughout reasoning. Experimental results demonstrate that TIM sustains high inference throughput, even when manipulating up to 90% of the key-value cache in GPU memory, and delivers accurate reasoning on mathematical tasks, effectively handling information retrieval challenges that require long-horizon reasoning and multi-hop tool use.

**Strengths:**

1. Introduction of a recursive and decompositional problem-solving method, improving structured reasoning.
2. Effective context pruning mechanism that supports virtually unlimited working memory and multi-hop tool calls.
3. High inference throughput maintained even with significant manipulation of GPU memory resources. Demonstrated accuracy in reasoning on complex tasks, including mathematical problem-solving and information retrieval.

**Weaknesses:**

1. Lack of detailed comparison with existing models to highlight relative performance improvements.
2. Insufficient discussion on potential computational overhead introduced by the context pruning mechanism.
3. Limited exploration of the model's scalability across different domains and tasks.

**Questions:**

1. How does TIM compare to existing models in terms of performance on standard benchmarks?
2. What are the computational costs associated with the context pruning mechanism, and how do they scale with model size?
3. Can TIM be effectively applied to tasks outside of mathematical reasoning and information retrieval?

---

### Meta-Review · Area_Chair_9tRC · 2026-01-07

**Summary:**

The reviewers' concerns fall into the following categories:
- **Limited, or even potentially flawed evaluation.** Models, tasks, baselines, and metrics considered are limited. Especially, the current benchmarks are not appropriate for evaluating long-context capabilities, which is the main focus of the paper. Furthermore, the performance of the base model is lower than what has been reported in prior works.
- **Lacks in-depth discussion.** No discussion regarding any computational overhead, scalability has been provided. A reviewer has also pointed out more ablation studies should have been provided.
The reviewers did not respond at all, and thus the concerns remain unchallenged. Hence, I recommend rejection.

**Reviewer Concerns:**

As the authors did not respond, all concerns are still outstanding.

**Reviewer Scores:**

None of the reviewers may have raised the score.

---

### Decision · Program_Chairs · 2026-01-26

Reject